# Evaluation of Fine Needle Aspiration Cytopathology in Salivary Gland Tumors under Milan System: Challenges, Misdiagnosis Rates, and Clinical Recommendations

**DOI:** 10.3390/biomedicines11071973

**Published:** 2023-07-12

**Authors:** Yi-Tien Huang, Chen-Yu Ho, Chun-Yen Ou, Cheng-Chih Huang, Wei-Ting Lee, Shu-Wei Tsai, Heng-Jui Hsu, David Shang-Yu Hung, Chien-Sheng Tsai, Sheen-Yie Fang, Sen-Tien Tsai, Jenn-Ren Hsiao, Chan-Chi Chang, Chien-Chin Chen

**Affiliations:** 1Department of Otolaryngology, National Cheng Kung University Hospital, College of Medicine, National Cheng Kung University, Tainan 704, Taiwan; bingoamazed@gmail.com (Y.-T.H.); hcychris@hotmail.com (C.-Y.H.); ojy1@mail.ncku.edu.tw (C.-Y.O.); cs841063@yahoo.com.tw (C.-C.H.); wendelllee92@yahoo.com.tw (W.-T.L.); tsaisuwei@gmail.com (S.-W.T.); how8079@hotmail.com (H.-J.H.); hungdavidda10703@gmail.com (D.S.-Y.H.); crazysoul081@gmail.com (C.-S.T.); sheen@mail.ncku.edu.tw (S.-Y.F.); t602511@mail.ncku.edu.tw (S.-T.T.); hsiaojr@mail.ncku.edu.tw (J.-R.H.); 2Department of Pathology, Ditmanson Medical Foundation Chia-Yi Christian Hospital, Chiayi 600, Taiwan; 3Department of Cosmetic Science, Chia Nan University of Pharmacy and Science, Tainan 717, Taiwan; 4Ph.D. Program in Translational Medicine, Rong Hsing Research Center for Translational Medicine, National Chung Hsing University, Taichung 402, Taiwan; 5Department of Biotechnology and Bioindustry Sciences, College of Bioscience and Biotechnology, National Cheng Kung University, Tainan 701, Taiwan

**Keywords:** benign neoplasm, cytopathology, fine-needle aspiration, FNA, Milan system, MSRSGC, pleomorphic adenoma, salivary gland, Warthin’s tumor

## Abstract

(1) Background: Salivary gland tumors are rare in the head and neck. To determine the need and extent of surgical intervention, fine needle aspiration (FNA) is a widely accepted tool to approach salivary gland lesions. However, the FNA cytology varies between entities, while the lack of uniform terminology makes diagnosis more challenging. Since establishing the Milan system for reporting salivary gland cytopathology (MSRSGC) has become an increasingly accepted reporting standard, further examination and detailed recommendations were needed. (2) Methods: Between April 2013 and October 2021, 375 cases with FNA and salivary gland resection were retrospectively collected. All FNA specimens were reclassified according to the criteria of MSRSGC. After surgical excision, the FNA data were compared with the histological diagnosis to estimate the risk of malignancy (ROM), the risk of neoplasm (RON), and the diagnostic accuracy for each diagnostic category. (3) Results: Our cohort’s distribution of ROM and RON was similar to the MSRSGC’s recommendation. Carcinoma ex pleomorphic adenoma (CXPA) has the highest rate (66.7%) of misdiagnosed as a nonneoplastic lesion or benign salivary gland tumor. Pleomorphic adenoma (PA) and Warthin’s tumor were the most common benign salivary gland tumors, while the cytology diagnosis of Warthin’s tumor seems more challenging than PAs. (4) Conclusions: Despite the convenience and effectiveness of MSRSGC, we suggest close follow-up, re-biopsy, or surgical removal for salivary lesions even in Milan IVA-Benign for possibly missing FNA of malignancy, mixed lesions, or prevention of malignant transformation.

## 1. Introduction

Salivary gland tumors are rare tumors that account for 3–6% of head and neck tumors [1,2,3]. Of these, about 14–27% are malignant [4,5]. Salivary gland tumors are composed of different tissue types and are classified into 36 distinct entities according to the 2022 World Health Organization (WHO) classification [6]. Due to their rarity and diversity, these types are challenging to characterize clinically and differentiate by imaging [1,3,6]. Ultrasound is a widely accepted and easily accessible examination tool that can identify ill-defined tumor margins, inhomogeneous echo structure, increased vascularity, and pathologic local lymph nodes in superficial salivary gland lesions. However, it is inadequate for low-grade malignancies or deep salivary gland structures, resulting in a pooled sensitivity of 66% and specificity of 92% [7,8]. Computerized tomography (CT) is beneficial for its detailed imaging and can provide more information about tumors, such as local bone or vascular invasion. However, it is limited by radiation dose considerations and the side effects of contrast agents. In recent years, CT can be combined with PET for a more accurate diagnosis of salivary gland tumors, including distant lung and bone and lymph node metastases, impacting subsequent treatment choices. Multiparametric magnetic resonance imaging (MRI) can provide more accurate soft tissue resolution. It can show a glandular or extra-glandular spread of salivary gland lesions, the perineural spread (including facial and trigeminal nerves), invasion of some cortical bony structures, and even intracranial structures. In the literature, MRI has a pooled sensitivity of 80% and specificity of 90% [7,8]. However, its disadvantages include high costs and MRI-related contraindications.

Clinically, ultrasonography combined with fine needle aspiration (FNA) is important in initial diagnosis due to its convenience, economy, and low invasiveness [9,10,11,12,13]. The 2021 American Society of Clinical Oncology (ASCO) guideline also recommends FNA or core needle biopsy (CNB) as a preoperative evaluation for salivary gland lesions [3]. Compared to CNB, FNA is less likely to cause pain, bleeding, subcutaneous hematoma, or even tumor seeding complications and does not require anesthesia, reducing the risk of anesthesia. However, due to insufficient sample volume, FNA is more likely to result in unreadable or misdiagnosed cases [14,15]. Therefore, detailed cytological reports and uniform classification criteria are extremely important in distinguishing malignant tumor risks and deciding surgical options. Although different scholars proposed their reporting systems, such as Bajwa et al. [16] and Wang et al. [17], FNA cytological reports still lack uniform reporting vocabulary and classification, making it easy for pathologists and surgeons to be confused or misunderstood. Therefore, as a result of international collaborative efforts, the Milan system for reporting salivary gland cytopathology (MSRSGC), which was established in Milan in 2015, has become an increasingly accepted reporting standard and clinical recommendation [18,19]. However, MSRSGC is mainly used in Europe and America, and only a few institutions in Asia have published related studies, such as Hirata et al. [20] and Wu et al. [21]. Most reports were from pathologists, with few clinical otolaryngologists and interventional radiologists sharing related experiences.

Clinical physicians seemed more concerned with effective treatment options, surgical options, and the avoidance of complications. For example, in the surgical removal of salivary lesions, according to the updated National Comprehensive Cancer Network (NCCN) guideline, all salivary gland tumors except for T4b are recommended to receive surgical treatment [22]. In parotid tumors, shallow lesions that do not invade the facial nerve do not require the sacrifice of the facial nerve, while facial nerve monitoring is required during surgery. Moreover, the low malignancy of the lesions can tolerate smaller margins, usually 1 cm, and reduce the risk of subsequent facial paralysis (8–38%) and Frey’s syndrome (20–40%) [23,24]. Minor salivary gland tumors have worse overall survival compared to major salivary gland tumors, and past reports also indicate that the negative margin of surgery significantly impacts prognosis [25]. In addition, cervical lymph node clearance may be necessary for patients with lymph node metastasis, and the associated complications include the marginal mandibular nerve, spinal accessory nerve, hypoglossal nerve, and vagus nerve injuries [23,24].

We believe that clinical doctors should have more interest and challenges in MSRSGC. After all, MSRSGC is designed to help clinical treatment guidelines. Moreover, using MSRSGC, even different pathologists can achieve high consistency in diagnosis and classification [26]. An effective and systematic reporting classification and clinical guidelines can affect treatment choice, reduce treatment complications, and improve patients’ quality of life. Therefore, considering that there are not many published articles about MSRSGC in Asia and from otolaryngologists, and most published articles did not have a consensus on the discussion of intermediate categories and subclassification, we aimed to examine the effectiveness of MSRSGC in the diagnosis of salivary gland lesions, discuss the cellular morphology and diagnostic problems of intermediate categories, and combine clinical experience to provide clinical perspectives.

## 2. Materials and Methods

This study was approved by the Institutional Review Board (IRB) of the National Cheng Kung University Hospital, Taiwan (IRB No. A-ER-110-391). Between April 2013 and October 2021, 375 cases with a salivary gland resection were retrospectively recruited. The clinicopathological characteristics, including sex, age, anatomical location of the lesion, FNA diagnosis, and surgical pathology diagnosis according to the fifth edition of the World Health Organization Classification of Head and Neck Tumors [27], were reviewed and confirmed again with two independent pathologists. In all cases, specimens from direct FNA smears were fixed in alcohol and stained with Papanicolaou stain.

This study cohort included consecutive benign or malignant epithelial salivary gland tumors, soft tissue lesions, and metastatic tumors to the salivary glands. Patients who received FNA more than a year before the resection were excluded because of the possible malignant transformation during this period and the uncertainty of the disease’s clinical course. The MSRSGC guidelines were used to retrospectively classify all cytopathological findings into an MSRSGC category. Specimens including absent interpretable lesional cells (<60), poor fixation, necrotic debris, excessive blood, and nonmucinous cyst fluid without an epithelial component were categorized as Milan I-nondiagnostic (ND). Cases with abscess/inflammation, reactive lymph nodes, and epithelial cysts were categorized as Milan II-nonneoplastic (NN). Cases with basaloid or oncocytic cells, mucinous cyst component, atypical lymphoid cells, or hypocellular specimens with atypical cells were categorized as Milan III-atypia of undetermined significance (AUS). Cases of pleomorphic adenoma (PA), Warthin tumor, hemangioma, lipoma, and schwannoma were categorized as Milan IVA-benign neoplasm (Benign). Basaloid neoplasms with or without matrix, cystic neoplasms with granular or vacuolated cytoplasm, and neoplasms with clear cells that did not have enough cellular material for ancillary studies to reach a definitive diagnosis were categorized as Milan IVB-salivary gland neoplasms of unknown malignant potential (SUMP). Cases with few but markedly atypical cells in correlation with clinical and imaging findings were categorized as Milan V-suspicious for malignancy (SM). Cases that were diagnostic of malignancy were categorized as Milan VI-malignant (M).

After surgical excision, the FNA data were compared with the histological diagnosis to estimate the risk of malignancy (ROM) and the risk of neoplasm (RON) for each diagnostic category. In addition, the diagnostic accuracy and false-negative rates of FNAC for benign and malignant salivary gland lesions have been analyzed. Among these, we have elaborated on the distribution of cytological diagnosis results for Milan IVA-Benign and Milan IVB-SUMP.

## 3. Results

### 3.1. The Reclassified Data According to the MSRSGC

A total of 375 unique cases were analyzed in this study. The demographics are shown in Table 1. The 375 cases, out of which 61.6% (*n* = 213) were males, and the rest were females 43.2% (*n* = 162) with a male: female ratio of 1.3:1. The mean age of the patients was 51.20 ± 14.83 years. These cases were reclassified according to the MSRSGC as follows: Milan I-ND, 61 (16.3%); Milan II-NN, 42 (11.2%); Milan III-AUS, 45 (12.0%); Milan IVA-Benign, 188 (50.1%); Milan IVB-SUMP, 23 (6.1%); Milan V-SM, 7 (1.9%); and Milan VI-M, 9 (2.4%). Of all FNA cytology, 335 aspirates were taken from the parotid gland, and the other 40 aspirates were taken from the submandibular gland. The highest ROM was found in the Milan VI-M (100.0%), followed by the Milan V-SM category (85.7%). In Table 2, the ROMs in our study are similar to the MSRSGC’s recommendation, while RONs in the nondiagnostic and nonneoplastic categories were higher than in other studies.

### 3.2. The False Negatives of Malignant Entities in Cytology

To determine the false negative rates of salivary gland FNAs for malignant entities, we chose the top ten prevalent malignant tumors to compare their false negative (Milan II-NN and Milan IVA-Benign) and true positive (Milan V-SM and Milan VI-M) results in cytology. Inconclusive (Milan III-AUS and Milan IVB-SUMP) and nondiagnostic (Milan I-ND) FNA diagnoses were taken as ambiguous. As seen in Figure 1, mucoepidermoid carcinoma (MEC) was the most prevalent malignancy subtype in our study (*n* = 14). Among all cases histologically diagnosed as malignancy, carcinoma ex pleomorphic adenoma (CXPA) has the highest rate of falsely diagnosed as Milan II-NN and Milan IVA-Benign (false negative rate: 66.7%), followed by acinic cell carcinoma (ACC) (28.6%), adenoid cystic carcinoma (AdCC) (25.0%), MEC (21.4%) and secretory carcinoma (20.0%). On the contrary, salivary duct carcinoma (SDC), adenocarcinoma not otherwise specified (AC-NOS), lymphoma, and metastatic carcinoma have the highest accuracy in FNA cytology diagnosis under the MSRSGC framework, 100% diagnosed as Milan V-SM or Milan VI-M—followed by squamous cell carcinoma (66.7%) and secretory carcinoma (40%), respectively.

### 3.3. ROM in Milan IVB-SUMP Category

The Milan IVB-SUMP constituted 6.1% (*n* = 23) of our case cohort. The histology results for the cases with Milan IVB-SUMP diagnoses of salivary gland FNAs showed that benign tumors made up 60.9% of the cases (*n* = 14). The most frequent benign entity is Warthin’s tumor (35.7%, *n* = 5). On the other hand, malignant tumors made up 34.8% (*n* = 8), with low-grade MEC (25.0%, *n* = 2) and AdCC (37.5%, *n* = 3) being the two most common types. Figure 2 illustrates the histological distribution of the Milan IVB-SUMP.

### 3.4. Benign Neoplasms Have a Diverse Distribution with a High Accuracy

Histologically proven benign neoplasms (*n* = 375) were analyzed, including PA (43.2%, *n* = 162) and Warthin’s tumor (30.4%, *n* = 114). As highlighted in Figure 3A, the locations of PAs consisted of the parotid (82.1%) and submandibular gland (17.9%), while most PAs (78.4%) were cytologically classified as Milan IVA-Benign during the FNAs. On the other hand, among 131 cytologically favored PA in the FNA, only six of them were histologically diagnosed as non-PA, with four cases (3.0%) ultimately diagnosed as malignant tumors, including 2 AdCC (1 arising from PA), 1 ACC, and 1 CXPA. These results indicated that our FNA diagnosis accuracy for PA was 95.4%, with a ROM of 3.0%.

Meanwhile, all Warthin’s tumors were found in the parotid gland in this study, as shown in Figure 3B, and their FNA diagnoses showed a different distribution compared to PAs. Most FNA diagnoses of Warthin’s tumors were categorized as Milan IVA- Benign (43.6%, *n* = 50), followed by Milan I-ND (19.3%, *n* = 22), Milan II-NN (17.5%, *n* = 20), and Milan III-AUS (14.0%, *n* = 16) (Figure 3B). Warthin’s tumors had much higher percentages of Milan III-AUS than PA. On the other hand, among 55 cases that cytologically favored Warthin’s tumor during FNA, five were histologically proven non-Warthin’s tumors, with two (3.6%) malignant tumors (1 ACC and 1 MEC). Subsequently, FNA diagnosis accuracy for Warthin’s tumor was 90.9%, with a ROM of 3.6%. In addiction, one cytologically Milan II-NN case showed Warthin’s tumor in conjunction with CXPA after resection.

## 4. Discussion

FNA has become a standard technique for examining salivary gland cancers during the initial approach [32]. Although FNA is convenient and practically effective for high-grade or low-grade malignancies, it still poses a challenge for differentiating subtypes of certain salivary gland lesions due to its limited information. In the past, there was no unified reporting system or terminology, and inconsistent cellular classification standards often resulted in misunderstandings by clinical physicians and affected clinical judgments and treatment. Therefore, the appearance of the MSRSGC (an international classification standard) was necessary to classify salivary gland tumors based on initial FNA cytology according to their likelihood of malignancy and provide more specific clinical recommendations [19,33]. After the emergence of MSRSGC, many studies have confirmed its convenience and effectiveness in classification. Some studies have also raised their views on finer details, such as Cormier et al. believing that the ROM range in the Milan III-AUS and Milan IVB-SUMP categories is too large and needs to be more finely divided [34], Allison et al. emphasizing that oncocytoid and basaloid neoplasms have statistically significant ROM differences in the Milan IVB-SUMP category [35], and Chowsilpa et al. believing that immunohistochemical staining can assist in the determination of Milan IVB-SUMP classification types [36]. These discussions affect the future of MSRSGC and whether it will provide a more confident and consistent bridge of cooperation between pathology and clinical physicians.

In our results, percentages in the three categories of Milan I-ND, Milan II-NN, and Milan III-AUS were higher than those reported by other groups (Table 2), which may be related to Taiwan’s unique medical system. For example, most patients with salivary gland tumors are more likely to visit the hospital for an FNA examination due to the high medical accessibility and affordable costs in Taiwan. In addition, due to the coverage from Taiwan’s national health insurance system, individuals are more likely to undergo surgical excision instead of small-volume biopsies, increasing the number of benign lesions in our data. All the abovementioned also explain why Milan I-ND and Milan II-NN had a greater RON. High-grade malignant tumors, such as SCC, SDC, AC-NOS, lymphoma, and metastatic tumor, had a much lower false negative rate due to their noticeable cytological features (Figure 1); in contrast, some well-differentiated/low-grade tumors, such as ACC and AdCC, were much challenging in distinguishing the cytological specimens. For example, in Figure 1, except for high-grade malignant tumors (SCC, SDC, AC-NOS, lymphoma, and metastatic tumor), most malignant tumors could not be clearly classified as malignant or benign under FNA cytology, resulting in diverse judgment and high error rates. In addition, the malignant transformation from benign tumors, e.g., CXPA, may be misdiagnosed because the malignant component was not sampled or detected.

Interestingly, PAs accounted for 78.4% of benign tumors in our data. Notably, 72.4% of PAs were classified as Milan IVA-Benign under FNA, and none were classified as Milan V-SM or Milan VI-M. It can be seen that PA is a salivary gland lesion with a high prevalence rate and relatively prominent cell characteristics. However, in 131 FNA cases identified as PA, 4 were histologically malignant, including 2 AdCC (1 arising from PA), 1 ACC, and 1 CXPA. Overall, the diagnosis of PA under FNA was 95.4% accurate.

In contrast, the diagnostic accuracy of Warthin’s tumor seems more challenging. Only 43.9% of Warthin’s tumors were classified as Milan IVA-Benign under FNA, 19.3% were classified as Milan I-ND, Milan II-NN 17.5%, Milan III-AUS 14.0%, Milan IVB-SUMP 4.4%, and Milan V-SM 0.9%. The possible reason may be due to the oncytic morphology and lymphoid aggregates. Among all 55 patients diagnosed with Warthin’s tumor under FNA, 2 were histologically diagnosed as malignant tumors after surgery, including 1 ACC and 1 MEC. The accuracy rate of diagnosing Warthin’s tumor through FNA is 90.9%. Overall, the ROMs for distinguishing PA and Warthin’s tumor in FNA are 3.0% and 3.6%, respectively. These results are similar to the findings of Tommola et al. [37] and Fisher et al. [38].

In Milan IVA-Benign, the ROM for distinguishing PA is 2.3%, while that for distinguishing Warthin’s tumor is 3.9%. In Milan IVB-SUMP, we found that the ROM for distinguishing PA was 33.3% (3 out of 9), including AdCC, ACC, and secretory carcinoma. On the other hand, the ROM in Milan IVB-SUMP for distinguishing Warthin’s tumor was 50.0% (1 out of 2), while the malignant tumor is extranodal marginal zone B-cell lymphoma. Additionally, in Milan IVB-SUMP, the ROM for basaloid cell neoplasms was 33.3% (1 out of 3; the malignant one is AdCC), while that for oncocytic tumors was 33.3% (3 out of 9; the malignant tumors were two ACC and one secretory carcinoma). Since the case numbers for different subtypes are small, the ROMs of different subgroups may not be representative.

Compared to MSRSGC’s given ROM of 35% for SUMP, the ROM for Milan IVB-SUMP in this study was 34%. Regardless of whether it is a basaloid cell neoplasm or an oncocytic tumor, their ROMs are similar to the overall ROM. While Warthin’s tumor is an important differential diagnosis for both benign neoplasm and Milan IVB-SUMP, it has a higher ROM than PA in Milan IVB-SUMP, consistent with Allison et al.’s findings [35]. Therefore, although Warthin’s tumor is usually classified as a Milan IVA-Bening or Milan IVB-SUMP, it has a stronger indication for surgery rather than observation in Milan IVB-SUMP.

In previous studies, salivary gland tumors were malignant in 27.4% and benign in 72.6% [1]. In our data, most of the salivary glands were benign tumors. Benign salivary gland tumors may not require surgical resection, but long-term follow-up is required. However, some patients may lose follow-up during the tracking process. While 10% of PA may develop malignant transformation in 20 years [39,40], they behave aggressively, with poor 5-year survival [39]. Regardless of missing FNA of malignancy, mixed lesions, or prevention of malignant transformation, we suggested that practitioners should be vigilant even when benign neoplasms were reported through FNA. Without contraindications, those patients may need close follow-up, re-biopsy, or surgical removal.

## 5. Limitations

This retrospective study has many limitations. First, all cases were collected after surgical tumor excision, and a possible selection bias might not be avoidable. Second, all FNA techniques followed traditional protocols for Papanicolaou stain without applying cell blocks. This limitation may affect our diagnostic accuracy since cell blocks can be performed for ancillary tests. Third, this is a single-institute study with limited retrospective data. More comprehensively designed, prospective, multi-center studies are needed to validate the effectiveness and practical usefulness of MSRSGC.

## 6. Conclusions

The MSRSGC is an excellent reporting method for salivary gland tumors that gives valuable guidance for evaluating cancer risk. FNA may misdiagnose certain low-grade malignant tumors. However, for benign salivary gland tumors, it is recommended to maintain long-term follow-up, even with re-biopsy or surgical therapy, to limit the risk of misdiagnosis or malignant change. It is important to note that MSRSGC is an effective system for reporting benign or malignant salivary gland tumors. The definitive diagnosis, however, should be made by a team consisting of pathologists, radiologists, and surgeons who consider the patient’s clinical presentation, imaging characteristics, and medical history.

## Figures and Tables

**Figure 1 biomedicines-11-01973-f001:**
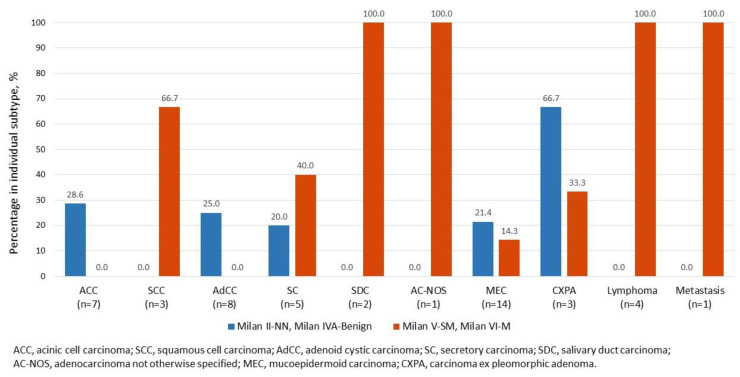
The discrepancy between fine needle aspiration (FNA) diagnosis and histology in different malignant salivary gland tumors. Milan II-NN and Milan IVA-Benign were defined as negative results, while Milan V-SM and Milan VI-M were positive. Abbreviations: Milan II-nonneoplastic (NN), Milan III-atypia of undetermined significance (AUS), Milan IVA-benign neoplasm (Benign), Milan IVB-salivary gland neoplasms of unknown malignant potential (SUMP), Milan V-suspicious for malignancy (SM), Milan VI-malignant (M), acinic cell carcinoma (ACC), squamous cell carcinoma (SCC), adenoid cystic carcinoma (AdCC), secretory carcinoma (SC), salivary duct carcinoma (SDC), adenocarcinoma not otherwise specified (AC-NOS), mucoepidermoid carcinoma (MEC), carcinoma ex pleomorphic adenoma (CXPA).

**Figure 2 biomedicines-11-01973-f002:**
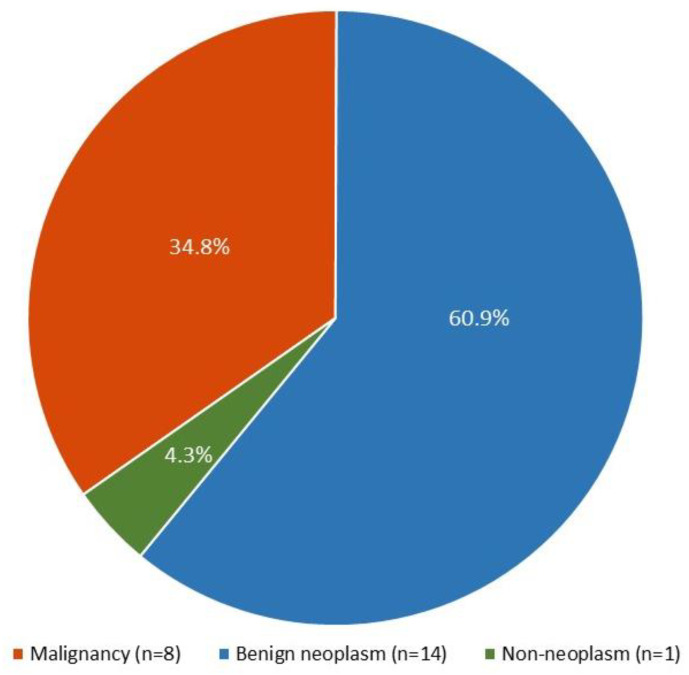
The histology distribution for the Milan IVB-SUMP of salivary gland FNA.

**Figure 3 biomedicines-11-01973-f003:**
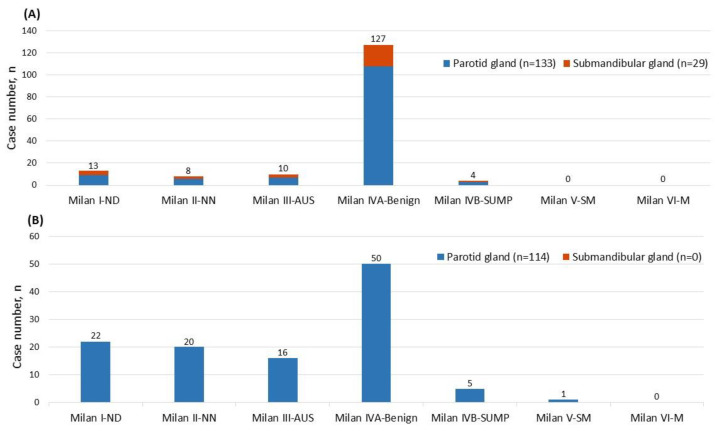
(**A**) Distribution of pleomorphic adenoma among the Milan system for reporting salivary gland cytopathology diagnostic categories (MSRSGC) and its anatomical location; (**B**) Distribution of Warthin’s tumor among MSRSGC diagnostic categories and its anatomical location.

**Table 1 biomedicines-11-01973-t001:** MSRSGC diagnostic categories and their ROMs and RONs in this study.

	Milan I-ND	Milan II-NN	Milan III-AUS	Milan IVA-Benign	Milan IVB-SUMP	Milan V-SM	Milan VI-M	Total
Total number, *n* (%)	61 (16.3)	42 (11.2)	45 (12.0)	188 (50.1)	23 (6.1)	7 (1.9)	9 (2.4)	375 (100.0)
Gender, *n* (%)								
Male	39 (63.9)	29 (69.0)	24 (53.3)	100 (53.2)	11 (47.8)	3 (42.9)	7 (77.8)	213 (56.8)
Female	22 (36.1)	13 (31.0)	21 (46.7)	88 (46.8)	12 (52.2)	4 (57.1)	2 (22.2)	162 (43.2)
Age, M (SD)	52.6 (14.3)	53.3 (14.0)	52.6 (7.8)	49.4 (15.2)	52.0 (24.0)	54.4 (12.9)	59.8 (15.1)	51.2 (14.8)
Location (%)								
Parotid gland	54 (88.5)	37 (88.1)	40 (88.9)	169 (89.9)	21 (91.3)	7 (100.0)	7 (77.8)	335 (89.3)
Submandibular gland	7 (11.5)	5 (11.9)	5 (11.1)	19 (10.1)	2 (8.7)	0 (0.0)	2 (22.2)	40 (10.7)
Malignancy, *n* (ROM, %)	10 (16.4)	4 (9.5)	11 (24.4)	5 (2.7)	8 (34.8)	6 (85.7)	9 (100.0)	56 (14.9)
Neoplasm, *n* (RON, %)	53 (86.9)	35 (83.3)	44 (97.8)	187 (99.5)	22 (95.6)	7 (100.0)	9 (100.0)	361 (96.3)

Abbreviations: MSRSGC, Milan system for reporting salivary gland cytopathology; ROM, risk of malignancy; RON, risk of neoplasm; Milan I-nondiagnostic (ND); Milan II-nonneoplastic (NN); Milan III-atypia of undetermined significance (AUS); Milan IVA-benign neoplasm(Benign); Milan IVB-salivary gland neoplasms of unknown malignant potential (SUMP); Milan V-suspicious for malignancy (SM); Milan VI-malignant (M).

**Table 2 biomedicines-11-01973-t002:** The comparison for MSRSGC diagnostic categories, ROMs, and RONs among different cohorts.

	Milan I-ND	Milan II-NN	Milan III-AUS	Milan IVA-Benign	Milan IVB-SUMP	Milan V-SM	Milan VI-M	Total
The current study (*n*, %)	61 (16.3)	42 (11.2)	45 (12.0)	188 (50.1)	23 (6.1)	7 (1.9)	9 (2.4)	375 (100.0)
ROM (*n*, %)	10 (16.4)	4 (9.5)	11 (24.4)	5 (2.7)	8 (34.8)	6 (85.7)	9 (100.0)	56 (14.9)
RON (*n*, %)	53 (86.9)	35 (83.3)	44 (97.8)	187 (99.5)	22 (95.6)	7 (100.0)	9 (100.0)	361 (96.3)
Jalaly, 2020 (*n*, %) [28]	871 (10.3)	875 (10.3)	610 (7.2)	3589 (42.3)	962 (11.3)	317 (3.7)	1269 (14.9)	8493 (100.0)
ROM (*n*, %)	147 (16.9)	92 (10.5)	240 (39.3)	105 (2.9)	379 (39.4)	267 (84.2)	1237 (97.5)	2467 (29.0)
RON (*n*, %)	475 (54.5)	161 (18.4)	469 (76.9)	3578 (99.7)	926 (96.3)	306 (24.1)	1263 (99.5)	7178 (84.5)
Hang, 2018 (*n*, %) [29]	70 (10.1)	80 (11.5)	40 (5.6)	315 (45.4)	59 (8.5)	15 (2.2)	115 (16.6)	694 (100.0)
ROM (*n*, %)	12 (17.1)	8 (10.0)	15 (37.5)	9 (2.9)	24 (40.0)	15 (100.0)	113 (98.3)	196 (28.2)
RON (*n*, %)	NA	NA	NA	NA	NA	NA	NA	NA
Song, 2019 (*n*, %) [30]	45 (10.5)	14 (3.3)	49 (11.4)	178 (41.5)	56 (13.8)	19 (44.3)	68 (15.9)	429 (100.0)
ROM (*n*, %)	8 (17.8)	2 (14.3)	15 (30.6)	4 (2.2)	26 (46.4)	15 (79.8)	67 (98.5)	137 (31.9)
RON (*n*, %)	29 (64.4)	3 (21.4)	39 (79.6)	178 (100.0)	56 (100.0)	18 (94.7)	68 (100)	391 (91.1)
Dubucs, 2019 (*n*, %) [31]	47 (21.8)	9 (4.2)	2 (1.0)	105 (48.6)	11 (5.1)	16 (7.4)	26 (12.6)	216 (100.0)
ROM (*n*, %)	16 (34.2)	0 (0.0)	0 (0.0)	4 (3.1)	5 (45.5)	11 (68.8)	26 (100.0)	62 (28.7)
RON (*n*, %)	38 (80.9)	1 (11.1)	2 (100.0)	105 (100.0)	7 (63.6)	13 (81.3)	26 (100.0)	198 (91.7)
MSRSGC								
ROM (%)	25	10	20	<5	35	60	90	NA

Note: The case numbers in all studies were limited to those with histological confirmation, and cases without histology were excluded. NA, not applicable.

## Data Availability

The raw data are unavailable due to privacy and ethical restrictions.

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
