# Peer review of "Evaluation of Fine Needle Aspiration Cytopathology in Salivary Gland Tumors under Milan System: Challenges, Misdiagnosis Rates, and Clinical Recommendations"

_biomedicines, 2023, doi:10.3390/biomedicines11071973_

Round 1
Reviewer 1 Report
Manuscript ID: biomedicines-2424049
Title: Evaluation of Fine Needle Aspiration Cytopathology in Salivary Gland
Tumors under Milan System: Challenges, Misdiagnosis Rates, and Clinical
Recommendations.
The purpose of this research is to classify salivary gland tumors using the Milan system, and we agree that this is a very promising and challenging attempt.
We believe this paper represents a milestone in the classification of salivary gland tumors.
However, this study has room for improvement or additional cases needed. In other words, the overall number of cases of malignant tumors of the salivary glands is small.
Please add at least 3~5 more cases to the number of cases in Figure 1, because the reason is insufficient for statistical analysis.
umors under Milan System: Challenges, Misdiagnosis Rates, and Clinical
Recommendations.
The purpose of this research is to classify salivary gland tumors using the Milan system, and we agree that this is a very promising and challenging attempt.
We believe this paper represents a milestone in the classification of salivary gland tumors.
However, this study has room for improvement or additional cases needed. In other words, the overall number of cases of malignant tumors of the salivary glands is small.
Please add at least 3~5 more cases to the number of cases in Figure 1, because the reason is insufficient for statistical analysis.
Author Response
Response letter to reviewer 1
Reviewer 1:
- The purpose of this research is to classify salivary gland tumors using the Milan system, and we agree that this is a very promising and challenging attempt. We believe this paper represents a milestone in the classification of salivary gland tumors.
Reply:
Thank you for your kindness and professional comments.
- However, this study has room for improvement or additional cases needed. In other words, the overall number of cases of malignant tumors of the salivary glands is small. Please add at least 3~5 more cases to the number of cases in Figure 1, because the reason is insufficient for statistical analysis.
Reply:
Thank you for your valuable comments on our manuscript. We appreciate your feedback and subsequently add malignant cases with metastatic squamous cell carcinoma (n=1) and primary lymphoma (n=4; 1 x follicular lymphoma, 2 x extranodal MALToma, 1 nodal MAToma) in Figure 1. In the original version, we just wanted to compare the benign (Milan-II and IVA) and malignant (Milan-V and VI) cytology diagnoses with their final histology among salivary gland tumors only. According to your advice, we put cases with metastatic carcinoma and primary lymphoma into comparison.

Reviewer 2 Report
This article correspond to a retrospective study on accuracy of FNA in salivary gland tumors using the Milan System and correlated to histologic results after surgery.
Specific comments
Page 3 line 124 : what do you mean for « revisions of FNA were excluded” ?
Page 4 table 2 : I understand that the reports indicated have taken into account only FNA verified after surgery. But for Jalaly 2020, which is a meta analysis of several articles I don’t understand the numbers that you indicate (n, ROM and RON)
Page 5, lines 177 and 181: the term MASC is no longer used, it has been replaced by secretory carcinoma
Page 5 Figure1: the legend does not correspond to the figure. Moreover the figure is difficult to understand. The results should not be reported as percentages because they do not clearly illustrate the proportion of each tumor. Why not indicate on the graph the tumors that were classified as SUMP? It would be better to see in what category the malignant tumor was initially reported.
Page 5 line 194: the figure 2 does not indicate the location of the tumor
Page 7 line 233: perhaps there is a “bias” in your study as only I-ND and II-NN verified after surgery have been included. In most studies ND et NN do not systematically require surgery if clinical and imaging data do not support a neoplasm
Page 7 line 249: it is surprising that only 44% of Warthin’s tumors have been reported as IVA-BN. Did the FNA were performed under US-assistance? As WT is a cystic tumor, the sampling needs to involve the wall of the cyst.
Page 7, lines 262-264: the ROM is evaluated on very small number of tumors, it may not be representative
Page 7, lines 243-270 : most data are not previously reported and should be inserted in the result section.
Author Response
Response letter to Reviewer 2
Reviewer 2:
1. This article correspond to a retrospective study on accuracy of FNA in salivary gland tumors using the Milan System and correlated to histologic results after surgery.
Reply: Thank you for your professional comments. All revised parts would be on yellow background in the revised draft.
2. Specific comments Page 3 line 124 : what do you mean for « revisions of FNA were excluded” ?
Reply: Thank you for your query. This sentence is wrong, but it is originally meant to describe the FNA diagnoses should be revised according to the updated Milan system. To avoid misunderstandings, we deleted this sentence. Thank you for your comments.
3. Page 4 table 2 : I understand that the reports indicated have taken into account only FNA verified after surgery. But for Jalaly 2020, which is a meta analysis of several articles I don’t understand the numbers that you indicate (n, ROM and RON).
Reply: Thank you for your professional comments. You really saved us with extreme thoughtfulness and expertise. The case numbers and RONs/ROMs were incorrectly cited in Table 2. We corrected these numbers and rechecked all references in Table 2.
4. Page 5, lines 177 and 181: the term MASC is no longer used, it has been replaced by secretory carcinoma
Reply: Thank you for your professional comments. We have replaced mammary analogue secretory carcinoma (MASC) with secretory carcinoma, according to the updated WHO classification.
5. Page 5 Figure1: the legend does not correspond to the figure. Moreover the figure is difficult to understand. The results should not be reported as percentages because they do not clearly illustrate the proportion of each tumor. Why not indicate on the graph the tumors that were classified as SUMP? It would be better to see in what category the malignant tumor was initially reported.
Reply: Thank you for your professional comments. To determine the false negative rates of salivary gland FNAs for malignant entities, we chose the ten most prevalent malignant tumors in our case series to compare their negative (Milan II-NN or Milan IVA-Benign) and positive (Milan V-SM or Milan VI-M) results in cytology. We defined Milan II-NN and Milan IVA-Benign as the negative result, while Milan V-SM and Milan VI-M are positive results. Reerds et al. used the same strategy to evaluate false-negative results, and they thought indeterminate and nondiagnostic FNA diagnoses lack clarity for the patient and the treating clinician (Cancer Cytopathol. 2022;130(3):189-194. doi:10.1002/cncy.22532). To make it clear, we revised our description in section 3.2. Moreover, we thank your professional suggestion and have added Figure 2 and section 3.3 to discuss cases with Milan IVB-SUMP.
6. Page 5 line 194: the figure 2 does not indicate the location of the tumor.
Reply: Thank you for your comments. The different colors in the figure indicate the tumor’s anatomical locations (parotid gland: blue; submandibular gland: orange). We agree the figure 3 (originally figure 2) might not be clearly illustrated adequate information, so we updated the figure 3.
7. Page 7 line 233: perhaps there is a “bias” in your study as only I-ND and II-NN verified after surgery have been included. In most studies ND et NN do not systematically require surgery if clinical and imaging data do not support a neoplasm
Reply: Thank you for your professional comments. We totally agree with your comments. Our case-cohort indeed has a selection bias. In most scenarios, Milan-ND and Milan-NN don’t need to have surgeries. However, due to the affordable costs, highly-accessible medical services, and 100% coverage of Taiwan’s national health insurance system, citizens are more likely to take surgical excision instead of small-volume biopsies, resulting in higher case numbers of benign lesions in our data compared to other studies. We have explained it in the same paragraph.
8. Page 7 line 249: it is surprising that only 44% of Warthin’s tumors have been reported as IVA-BN. Did the FNA were performed under US-assistance? As WT is a cystic tumor, the sampling needs to involve the wall of the cyst.
Reply: Thank you for your professional comments. All FNAs were performed under ultrasound guidance. However, most FNAs were done by junior resident doctors with monthly duty rotations. Honestly, most of our Warthin’s tumours have poor cellularity in FNA. Herein, we believe such data may be associated with poor aspiration quality and inexperienced practitioners.
9. Page 7, lines 262-264: the ROM is evaluated on very small number of tumors, it may not be representative.
Reply: We totally agree with you. The case number is small, and the ROMs for different subgroups may not be representative. We have added such comments to the discussion. Thank you for your professional comments.
10. Page 7, lines 243-270 : most data are not previously reported and should be inserted in the result section..
Reply: Thank you for your professional comments. We have incorporated them into results of sections 3.3 and 3.4.

Round 2
Reviewer 1 Report
Ms. Ref. No. biomedicines-2424049
Title: Evaluation of Fine Needle Aspiration Cytopathology in Salivary Gland Tumors under Milan System: Challenges, Misdiagnosis Rates, and Clinical Recommendations.
This paper is useful for Salivary Gland Tumors under Milan System. This paper should be considerable interest to many of the readers of biomedicines.
So, this paper should be published without change.
Ms. Ref. No. biomedicines-2424049
Title: Evaluation of Fine Needle Aspiration Cytopathology in Salivary Gland Tumors under Milan System: Challenges, Misdiagnosis Rates, and Clinical Recommendations.
This paper is useful for Salivary Gland Tumors under Milan System. This paper should be considerable interest to many of the readers of biomedicines.
So, this paper should be published without change.
Author Response
Reviewer 1:
- This paper is useful for Salivary Gland Tumors under Milan System. This paper should be considerable interest to many of the readers of biomedicines. So, this paper should be published without change.
Reply:
Thank you for your kindness and professional comments.

Reviewer 2 Report
Thank you for your corrections, the manuscript has been improved
I just have two comments :
page 4, table2, Jalaly study, the numbers for RON do not correspond to the original article (ROM are OK), they should be corrected
page 9, lines 313-314: I am not sure that the use of liquid-based cytology would improve the results. It may be true for cell blocks as they allow to perform immunocytochemistry and FISH, ancillaries studies that really improve the cytological diagnosis
Author Response
Response letter to Reviewer 2
Reviewer 2:
1. Thank you for your corrections, the manuscript has been improved
Reply: Thank you for your kindness.
2. I just have two comments :
Reply: Thank you for your professional comments.
3. page 4, table2, Jalaly study, the numbers for RON do not correspond to the original article (ROM are OK), they should be corrected.
Reply: Thank you for your thoughful comments. We truly appreciate your time and check. Because of your advice, we have very carefully checked all the numbers in the Jalaly study (a meta-analysis), and found their total case number with surgical follow-up was wrong (not 8468, but 8493). If we checked their Table 1, you could find three studies had wrong total numbers (Bhutani 2019; Chen 2019; Choy 2019), and subsequently, they got the wrong total.
In addition, we calculated the RON based on the Jalaly study’s Table 2 (A) (the pooled analysis, whether or not studies distinguished or separately reported nonneoplastic lesions and benign neoplasms). Each category was calculated by the total specimen number minus non-neoplastic specimen number, e.g. ND: 871-396=475. We have corrected RONs in the revised draft.
4. page 9, lines 313-314: I am not sure that the use of liquid-based cytology would improve the results. It may be true for cell blocks as they allow to perform immunocytochemistry and FISH, ancillaries studies that really improve the cytological diagnosis
Reply: Thank you for your professional comments. We appreciate your expertise and totally agree with your suggestion. We have revised the description according to your professional comments. All revised parts were on yellow background in the revised draft.
